# Objective perimetry identifies regional functional progression and recovery in mild Diabetic Macular Oedema

**Faran Sabeti**[1,2]*, **Bhim B. Rai**[1], **Josh P. van Kleef**[1], **Emilie M. F. Rohan**[1], **Corinne F. Carle**[1], **Richard C. Barry**[3,4], **Rohan W. Essex**[3,5], **Christopher J. Nolan**[3,5], **Ted Maddess**[1]

**1** The John Curtin School of Medical Research, Australian National University, Canberra, ACT, Australia, **2** Faculty of Health, School of Optometry, University of Canberra, Bruce, Canberra, Australia, **3** The Canberra Hospital, ACT Health, Garran, Canberra, ACT, Australia, **4** Blink Eye Clinic, Canberra, ACT, Australia, **5** ANU Medical School, Australian National University, Canberra, ACT, Australia

* faran.sabeti@anu.edu.au

## Abstract

### Purpose

Retinal function beyond foveal vision is not routinely examined in the clinical screening and management of diabetic retinopathy although growing evidence suggests it may precede structural changes. In this study we compare optical coherence tomography (OCT) based macular structure with function measured objectively with the ObjectiveFIELD Analyzer (OFA), and with Matrix perimetry. We did that longitudinally in Type 2 diabetes (T2D) patients with mild Diabetic Macular Oedema (DMO) with good vision and a similar number of T2D patients without DMO, to evaluate changes in retinal function more peripherally over the natural course of retinopathy.

### Methods

Both eyes of 16 T2D patients (65.0 ± 10.1, 10 females), 10 with baseline DMO, were followed for up longitudinally for 27 months providing 94 data sets. Vasculopathy was assessed by fundus photography. Retinopathy was graded using to Early Treatment of Diabetic Retinopathy Study (ETDRS) guidelines. Posterior-pole OCT quantified a 64-region/eye thickness grid. Retinal function was measured with 10–2 Matrix perimetry, and the FDA-cleared OFA. Two multifocal pupillographic objective perimetry (mfPOP) variants presented 44 stimuli/eye within either the central 30˚ or 60˚ of the visual field, providing sensitivities and delays for each test-region. OCT, Matrix and 30˚ OFA data were mapped to a common 44 region/eye grid allowing change over time to be compared at the same retinal regions.

### Results

In eyes that presented with DMO at baseline, mean retinal thickness reduced from 237 ± 25 µm to 234.2 ± 26.7 µm, while the initially non-DMO eyes significantly increased their mean thickness from 250.7 ± 24.4 µm to 255.7 ± 20.6 µm (both p<0.05). Eyes that reduced

**Data Availability Statement:** Data may not be publicly shared as a person's age, sex, and disease diagnosis and stage, years of diabetes, eyeglass use and prescription, prescription medication, past

eye surgery etc can be used to identify people with reasonable accuracy. The ANU Ethics Committee, local Health Department's Human Ethics Committee and ANU commercialisation have imposed this restriction on the dataset being shared. The ANU commercialisation office, would need to approve the data to be shared. They can be reached on commercialisation@anu.edu.au.

**Funding:** Diabetes Australia General Grant (2020) - FS https://www.diabetesaustralia.com.au/research-advocacy/diabetes-australia-research-program/ ANU PhD Scholarship - BBR ANU intramural Our Health in Our Hands - TM The funders had no role in study design, data collection and analysis, decision to publish, or preparation of the manuscript.

**Competing interests:** The authors have declared that no competing interests exist.

in retinal thickness over time recovered to more normal OFA sensitivities and delays (all p<0.021). Matrix perimetry quantified fewer regions that changed significantly over the 27 months, mostly presenting in the central 8 degrees.

## Conclusions

Changes in retinal function measured by OFA possibly offer greater power to monitor DMO over time than Matrix perimetry data.

## Introduction

Diabetic retinopathy, a structural and functional manifestation of proinflammatory activity within the retina in chronic hyperglycemia [1], is the leading cause of vision impairment in the working-aged population [2]. Diabetic macular oedema (DMO) is a primary vision threatening complication of diabetic retinopathy, with many studies supporting the role that anti-vascular endothelial growth factor (anti-VEGF) therapy plays in its management [3–5]. Possibilities for treatment with fenofibrate, a peroxisome proliferator activated receptor alpha agonist, used primarily to treat dyslipidemia and high -density lipoprotein cholesterol [6, 7] or other interventions are also emerging [8, 9]. This provides an impetus to improve our understanding of mild off-centre DMO so that recommendations on the initiation of management can be further developed.

It has been well established that diabetic retinal neurodegeneration (DRN) is a hallmark of early diabetic retinopathy and may precede vascular changes [10, 11]. Early cellular changes in the retina have been found to affect ganglion cell and inner plexiform layer structure leading to apoptosis [12, 13]. These early changes to the inner retina have led researchers to investigate functional impairment using electroretinographic recordings [11] and perimetry [14]. Whilst these functional measures have shown promise, electroretinography suffers from long setup times [15] and poor repeatability due to the subjective nature of testing and small stimulus sizes in standard automated perimetry [16].

Multifocal pupillographic objective perimetry (mfPOP) non-invasively assesses retinal function at many locations of both eyes concurrently. In addition, changes in both response amplitudes and response delays (implicit times) are measured at each region, across both retinas. In standard mfPOP methods found in the FDA-cleared ObjectiveFIELD Analyzer (OFA), forty-four regions/eye are concurrently tested within either the central 30° or 60°. Visual field maps of function are thus generated for both eyes at the same time, these include separate maps for per-region sensitivities and response delays. Thus, 4 visual field maps are generated in under 7 minutes and may be an advantage over forms of objective perimetry, such as multifocal electroretinography. In addition, OFA requires minimal training to administer with rapid setup times. Our lab has previously demonstrated the ability of mfPOP to accurately identify the clinical severity of diabetic retinopathy in early Type 2 Diabetes (T2D) [17, 18]. We have also shown topographical agreement between objective measures of DRN in the absence of retinopathy with multifocal visual evoked potentials (mfVEPs) and mfPOP abnormalities in T2D patients measured on the same day [19]. This confirms and builds evidence of the impact of diabetic neuropathy on the visual system at various stages of DR. Such advances in topological measures of retinal function [17, 18, 20, 21] may assist clinicians to identify early functional changes before the onset of new signs of DR [22]. Patients with diabetes may

also exhibit thinning of retinal nerve fibre layer [23, 24], and other layers prior to the onset of clinical DR [25, 26].

Although structural retinal changes evaluated by stereoscopic ophthalmoscopy remains the mainstay clinical assessment, managing DR becomes a reactionary procedure. One approach, which may circumvent progression to sight threatening forms of DR would be to identify early localised changes in the retina. We believe that changes in retinal function may represent the earliest signs of distress in the retina. The purpose of this study was to evaluate the ability of objective and subjective perimetry to detect longitudinal change in the natural course of DMO. We enrolled T2D patients with mild DMO, and similar T2D patients without DMO at baseline, and followed them longitudinally for 1.5 years. Given the presence of DMO our primary measure was macular retinal thickness as measured by Optical Coherence Tomography (OCT), and we did regression analyses to identify significant changes in regional thickness over the study period. Secondary measures were similar regression analyses of functional changes measured by Matrix per-region sensitivities, and OFA per-region sensitivities and delays. We also sought to examine the correlation between regional structural and functional changes and mapped the OCT, Matrix and OFA central 30˚ data (sensitives and delays) to a common 44-region grid [27]. Based on previous findings of retinal dysfunction early in DR, we hypothesize that the presence of peripheral changes in retinal function is associated with patients with off-centre-DMO and good vision. We seek to evaluate these changes utilising objective and subjective visual sensitivity measures to aid our understanding of the impact mild DMO may have on retinal function, thus guiding clinical monitoring and management of this condition.

## Methods

### Participants

This prospective study recruited sixteen T2D patients (10 females, 63.6 ± 8.01 y, mean ± SD), ten of whom presented at baseline with mild DMO in at least one eye. Subjects were identified following their DR screening by their attending clinician from private ophthalmology (RCB) and optometry (FS) practices in Canberra. Patients were then written to and invited to participate in the study. This was a convenience sample of persons who were at risk of progression to more serious retinopathy and their follow-up was determined by the protocol recommended by their ophthalmologist. Therefore, patients with DMO were followed up more frequently and received updates of their ocular health more often than they otherwise may have received. The frequency of examination feedback may have motivated some of the participants to improve their glycaemic control contributing to the resolution of DMO. In this light, examining Matrix and OFA functional data correlations with any changes in retinal thickness would be possible.

The inclusion criteria were LogMAR visual acuity < 0.3 and DMO patients had to have evidence of mild DMO, defined as retinal thickness ≥320 μm on spectral domain optical coherence tomography (OCT, Spectralis, Heidelberg Engineering, Germany) within the boundaries of the macula but not at the centre. The two primary clinicians, FS and BBR, had to agree that the inclusion criteria were met. Exclusion criteria included previous ophthalmic management of diabetic retinopathy, including laser or injections, and no evidence of other confounding vitreoretinal or optic nerve disease, including cataract surgery. Subjects were assessed at a median of 3 visits over 457 ± 198 days (mean ± SD) with a median interval of 243.5 ± 145.6 (median ± SD) between visits. The study endpoint was reached if a patient required treatment or at the conclusion of the study at 27 months. This study adhered to the tenets of the declaration of Helsinki and informed written consent was obtained by all participants prior to their

enrolment. It was approved by the Human Research Ethics Committees of the Australian Capital Territory health (Protocol ETH.10.13.291) and the ANU ethics committee (Protocol 2013/286).

## Examinations

At the baseline visit we measured logMAR BCVA with ETDRS chart 2 according to the threshold strategy published previously [28]. Next the patient's intra-ocular pressure (IOP) was measured with Goldmann applanation tonometry, and pachymetry was performed (Pachymate DGH 55, DGH technology Inc., USA). All subjects also completed Humphrey Matrix automated perimetry using the 10–2 program (Carl Zeiss Meditec, Dublin, CA, USA).

## Multifocal pupillographic objective perimetry

We measured objective visual field data with an FDA-cleared prototype of the ObjectiveFIELD Analyser (Konan Medical, Irvine, CA, USA). The OFA presents independent multifocal stimuli to both eyes, measuring direct and consensual amplitude and delay responses from each eye concurrently [17]. Response delays were represented by time from stimulus onset to peak contraction. Fig 1 illustrates that the two stimulus protocols comprised of yellow stimulus elements (regions) presented independently and concurrently to both eyes on two liquid crystal displays at optical infinity, with a 60 frame/s refresh rate. The stimulus array was presented in a dartboard layout (Fig 1) consisting of 44 regions, 11 in each quadrant, arranged in 5 rings. No regions crossed the horizontal or vertical meridians. Stimulus luminance was adjusted for each stimulus region to boost responses from inherently less sensitive regions producing a similar response amplitude profile across the field in healthy subjects [29]. The primary difference between the two stimulus variants were that the stimuli were scaled with eccentricity and subtended either ±15˚ or ± 30˚ from fixation (Fig 1) with a maximum luminance of 288 cd/m$^2$ presented on a 10 cd/m$^2$ dim yellow starburst background with a small central fixation cross. We refer to the two variants as P30˚ and P60˚ pertaining to the ±15˚ and ± 30˚ stimulus array, respectively. The International Commission on Illumination (CIE) x, y colour coordinates of the stimuli were (0.377, 0.464). The borders of the stimuli were blurred allowing tolerance of ametropia of ±3.0D. Binocular fusion was facilitated by presenting the dichoptic stimuli on a faint starburst background with a long thin white line along the vertical meridian, and by adjusting ocular vergence to the habitual phoria of the subject.

The individual stimuli appeared at random (33 ms duration) over time with a mean interval of 4 s/region, described previously as the Clustered Volleys method [30]. No overlapping regions (dashed lines) were activated simultaneously (Fig 1A, 1C). Total stimulus duration was 7–8 minutes divided into 9 segments of 40s with short rest breaks between segments. Blinks were detected in real-time and affected data were removed before analysis. If data loss exceeded 15% of the total collection the segment was repeated.

Pupil responses were measured as real-time pupil diameter, relative to the population mean pupil diameter of 3.5 mm. In addition, pupil neuropathy would produce a generalised depression of sensitivity or global delay in responses, unlike the focal per-region changes found in our studies of type 2 diabetes [17, 31], where per-region time to peak contraction delay can be significantly faster or slower than normal in the same eye [31]. Pupil diameter, three millimetres above the centre, was not recorded to minimise the effects of ptosis.

## Ocular exams

After OFA testing participants' pupils were dilated with 1.0% tropicamide and 2.5% phenylephrine. Evaluation of anterior and posterior segments were initially conducted by a retinal

## Stimulus Ensembles P60° and P30°

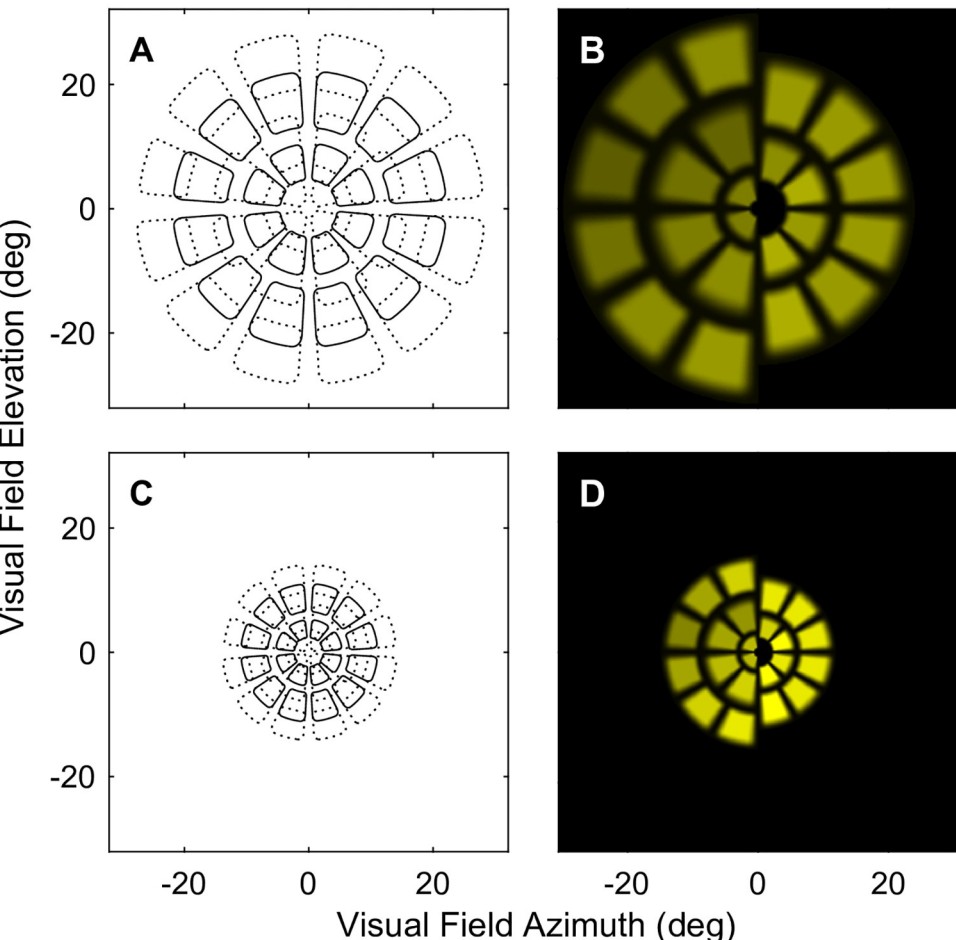

**Fig 1. Multifocal stimulus arrays for the P60˚ and P30˚ stimulus protocols.** The possible locations of the pseudo-randomly presented 44 test-stimuli/eye comprised five rings across: (A) the P60˚ array spanning ±30˚ from fixation; or (C) P30˚ array spanning ±15˚. Although stimuli could potentially overlap in space (dashed lines), in practice they were presented so that this never happened. Both P30˚ and P60˚ consisted of 5 rings of lozenge-shaped stimuli. Going from central to outer the borders of the rings are dashed (rings 1,3,5) or solid (rings 2,4). (B,D) show half the left-eye stimuli turned-on to Illustrate the luminance-balancing method that adjusts response sizes from normal subjects to be more even across the examined field. The two halves show rings 1,3,5 on the left and 2,4 on the right. There are thus 11 stimuli per quadrant. The maximum luminance of the P30˚ and P60˚ stimuli were 288 and 150 cd/m$^2$ respectively. The backgrounds were at 10 cd/m$^2$.

ophthalmologist (BBR) with slit-lamp biomicroscopy (BQ 900, Haag Streit, Switzerland). Subsequent digital imaging was conducted with OCT (see above). The circular optic-disc scan contained 768 A-scans and automatic real-time mean of 100 times and assessed the peripapillary retinal nerve fibre (RNFL) thickness. The macular scan was a dense central volume scan (12˚ × 12˚), centered on the fovea. 61 B-scans each spaced 120 μm apart, automatic real-time mean of 25 (768 A-scans). The data were reported as mean thickness within an 8×8 grid of square elements (Fig 2A) imaged with a 7-degree tilt to align with the position of the fovea to optic disc. Raw 8×8 grid data were then imported into MATLAB (2020b Mathworks Inc., Natick, MA) for analysis with customised code. That was followed by five 45˚ non-stereoscopic colour fundus photograph (Canon CR-2, Tokyo, Japan) equivalent to the standard 30˚, seven-

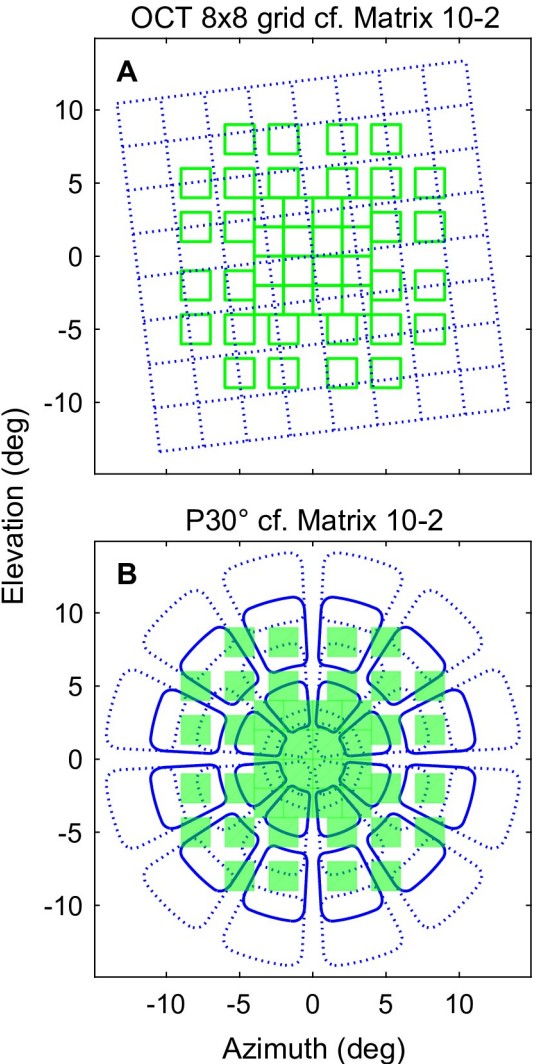

**Fig 2. (A)** Demonstration of the overlap between the optical coherence tomography (OCT) 8×8 thickness grid (blue) with the Matrix 10–2 stimulus array (green). Each OCT grid element is 3×3 deg, and each Matrix stimulus 2×2 deg. The retinal thickness data was flipped to represent the corresponding region in visual space. **(B)** The P30˚ stimulus array (blue) projected onto the Matrix 10–2 test grid (green). Weights based on overlapping areas were used to create 10–2 equivalent measures from the OCT and ObjectiveFIELD Analyzer (OFA) data. This allowed concordant region-by-region comparisons between the three data types.

image ETDRS standard [32]. Systemic blood pressure and glycosylated haemoglobin (HbA1c) values were recorded. Other ocular data was collected by an orthoptist (EMFR) and diabetic retinopathy diagnosis was classified by an ophthalmologist (BBR) according to the ETDRS guidelines [32].

## Analysis

A regressive method was used to estimate the pupil response for each region from the continuous pupil record [17, 33]. Per-region response amplitudes were measured in μm, AmpStd, and delays (times-to-peak constriction) in seconds, both relative to normal. Normative OFA data were based on a study of 133 participants each tested twice, 2 weeks apart [34]. For regression analysis we transformed the AmpStd μm to decibel sensitivity (dB) using a generalised log-

transform [33], which also stabilized the variance. The resulting per-region sensitivities and delays were estimated and analysed using MATLAB.

To examine the local spatial agreement between sensitivities or delays and changes in retinal thickness we remapped the OCT posterior pole 8×8 thickness grid data, and the P30˚ OFA data onto the Matrix 10–2 stimulus pattern (Fig 2A, 2B). Basically, the overlapping areas between the 10–2 and the OFA and OCT maps provided weights for combining the OFA and OCT data into 10–2 equivalent measures. Thus, the three data types could be compared region-by-region. We have previously published the mapping method in detail, and also the same mapping of the OCT grid onto 10–2 more recently [27]. We used multiple regression-linear models to examine the changes in retinal thickness, and changes in sensitivity and response delay overtime (see below). Thus, significant change was quantified by slopes with p-values $< 0.05$.

## Results

### Patient characteristics

Table 1 presents baseline characteristics across the sixteen subjects of the two study cohorts, T2D patients with DMO and those patients without DMO. Ten patients had off-centre DMO at baseline. There were no significant differences between the baseline demographic and ocular measurements. Mean diabetes duration, HbA1c and systolic blood pressure were greater in DMO subjects than non-DMO at baseline, these differences however, were not significant. Significant differences emerged in mean central retinal thickness at the end of the study between the DMO group (234.2 ± 26.7 μm) and the non-DMO group (255.7 ± 20.6 μm, $p = 0.023$). There was no significant difference in visual acuity at the end of the study relative to baseline.

**Table 1. Patient demographics.**

| Parameters | Non-DMO subjects ($N = 6$) | DMO subjects ($N = 10$) |
|---|---|---|
| *Values per subject*: Mean (±SD) | | |
| Age | 65.0 ± 10.1 | 62.2 ± 4.7 |
| Gender (%Males) | 67 | 40 |
| Diabetes Duration (y) | 6.75 ± 4.9 | 11.81 ± 6.6 |
| HbA1c (%) | 6.75 ± 0.8 | 7.6 ± 1.35 |
| Blood Pressure (mmHg) | | |
| Systolic | 132.1 ± 10.4 | 135.3 ± 13.5 |
| Diastolic | 81.3 ± 10.7 | 79.9 ± 11.3 |
| Matrix Mean Deviation (dB) | -1.19 ± 1.87 | -0.74 ± 2.40 |
| Central Retinal thickness (μm) | 250.7 ± 24.4 | 237.0 ± 25.0 |
| *Values per eye*: Mean (±SD) | | |
| Visual Acuity (LogMAR) | | |
| OD | 0.03 ± 0.12 | 0.08 ± 0.22 |
| OS | 0.01 ± 0.06 | 0.04 ± 0.1 |
| Intraocular Pressure (mmHg) | | |
| OD | 17.6 ± 2.9 | 16.1 ± 3.2 |
| OS | 17.7 ± 3.4 | 16 ± 3.0 |

Table presents mean ± SD of variables between two diabetes cohorts studied.

DMO = Diabetic Macular oedema; HbA1c = glycosylated haemoglobin. Only HbA1c was significantly different (t-test, p<0.05).

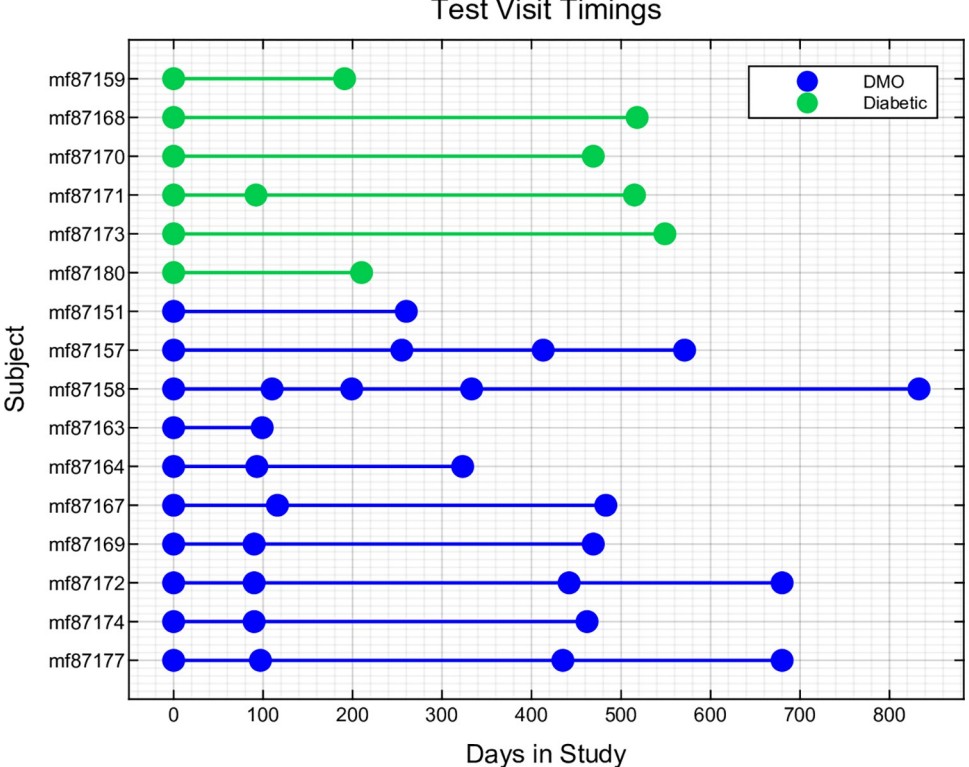

**Fig 3. The timing of the visits of the subjects, each solid circular dot represents a test visit.** As per the legend patients with diabetic macular oedema (DMO) at baseline are in blue, those without green.

Fig 3 shows the sequential examination schedule of each participant. Most DMO patients had a follow-up assessment within 3 months of their initial examination, and a minimum of 2 visits.

Fig 4 shows the results of the OCT retinal thickness data, the P30° OFA sensitivity and delay deviations from normal, and the Matrix total deviations (Fig 4A–4D and 4G–4J), all mapped to the 10–2 grid as per Fig 2. The last two rows (Fig 4E, 4F and 4K, 4L) show the sensitivity and delay results for the wide-field P60° stimulus, which cannot be mapped onto the 10–2 grid. The data are the outcomes for subject mf87157 on their first visit and their last visit 571 days (18.7 months) later. This subject was selected due to the clear corresponding changes between retinal thickness and function changes over time. The right eye retinal thickness was reduced. The P30° and P60° sensitivities improved while the delays appeared not to improve. The same data is presented with the baseline data for both eyes on the left in S2 Fig.

## Structure-function analysis

Fig 5 shows data from the 9 subjects who showed significant (p < 0.05) longitudinal changes (slopes) in per-region retinal function and structure presented on the 10–2 Matrix grid representing the central 10° of visual field. This allowed for a region-by-region comparison of OFA responses vs. Matrix and OCT retinal thickness. The longitudinal data over time were fitted by linear regression and those regions with significant change in retinal sensitivity or thickness over time between baseline and final visit are coloured. The colour-map legends at the bottom of the pairs of columns indicate the magnitude and sign of those per-region slopes. Notice that the data from four visits of mf87157 did as Fig 4 suggested, retinal thickness and sensitivities

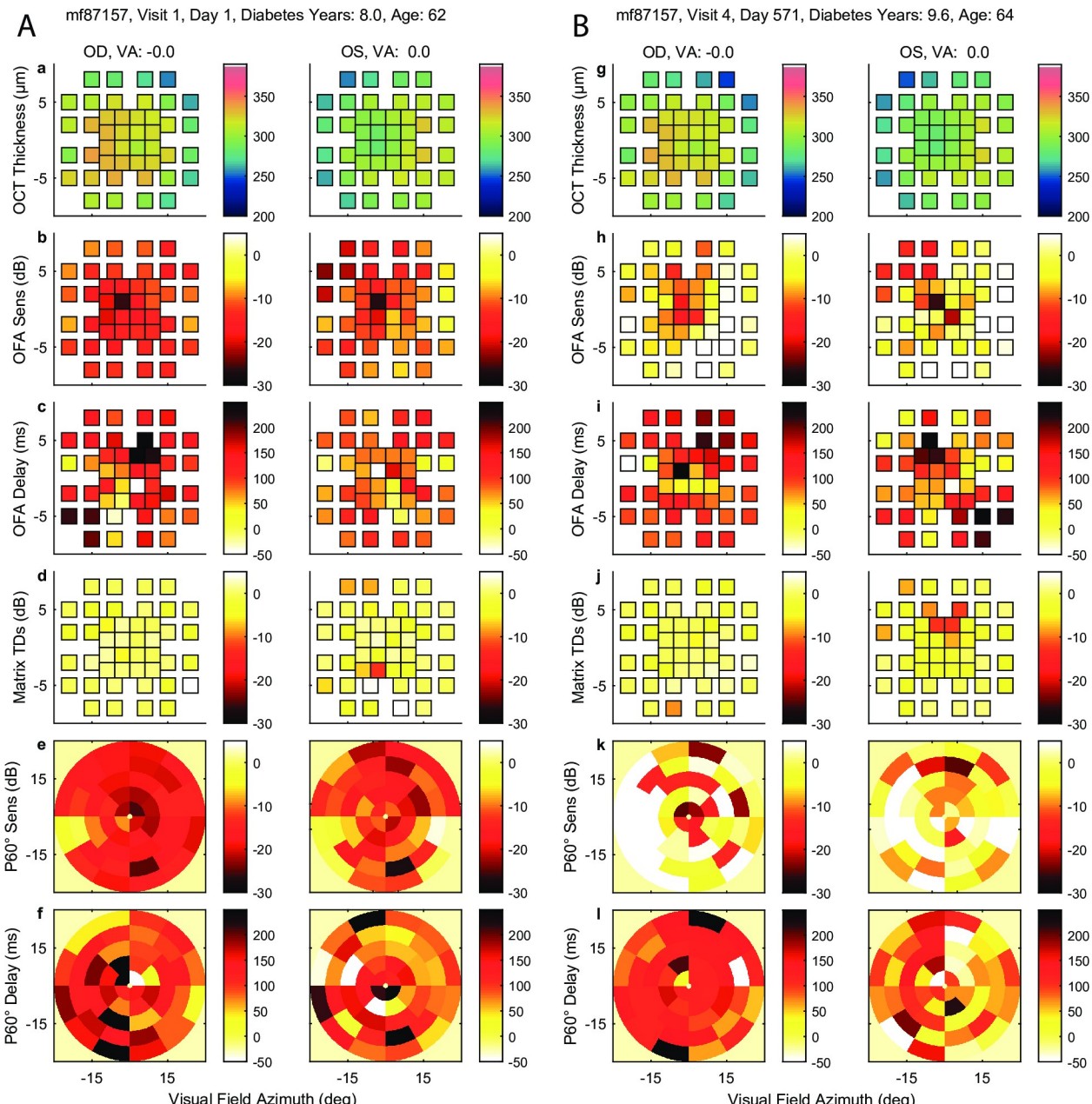

**Fig 4.** Illustration of data mapped to the 10–2 format (±10˚ eccentricity from fixation) for two visits 571 days apart (**A, B**) from a DMO patient whose age was 62 years at baseline. The pairs of columns in **A** (initial visit) and **B** (571 days later) are for OD and OS. From top to bottom the rows of data show: OCT retinal thickness, OFA sensitivities and delays from P30˚, Matrix Total Deviations (TDs), and the P60˚ OFA Sensitivities and Delays in their native polar format. The rows are identified by a to f for the first visit, and g to l for the second. The OCT data show reduction in thickness to more normal levels over the two visits, especially OD. The P30˚ and P60˚ sensitivities seem to follow the reduction in DMO with increased sensitivities. The delays seem to change less. Readers may wish to free-fuse A and B. See all 47 data sets, including for two visits between A and B, in the S1 Fig.

improved. Subjects mf87157, mf87164 and mf87167 (Fig 5 rows A, B D), who presented with DMO at baseline, demonstrated significantly reduced retinal thickness (blue regions) with corresponding gains in OFA sensitivity (green regions). Matrix sensitivity showed almost no significant change. Patient mf87171 (row I), a T2D patient without initial DMO showed

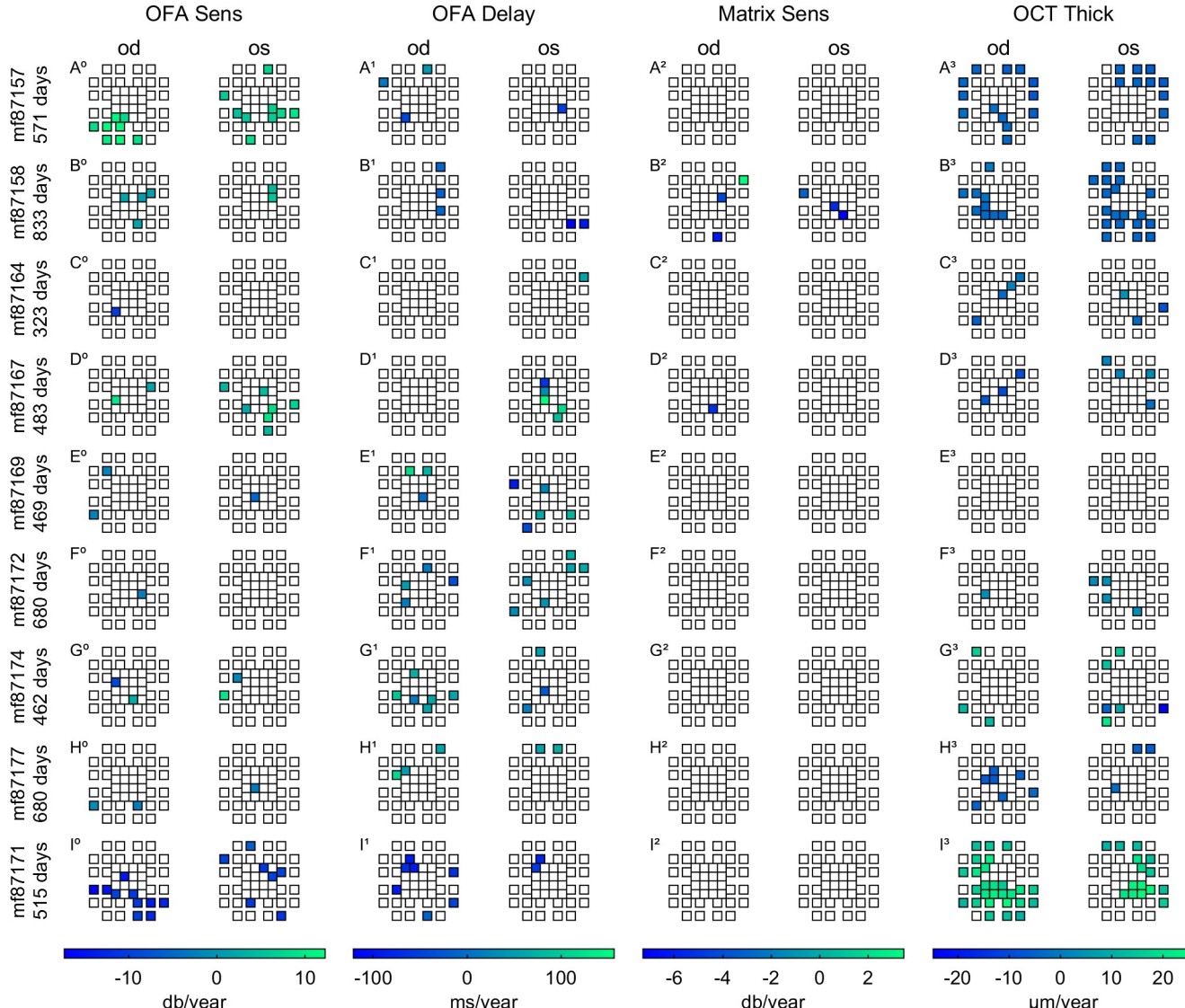

**Fig 5. Data for subjects with significant slopes (regional progression or recovery colour coded, non-significant regions white) for the 4 data types over the study period.** The only diabetic (non-DMO) person with significant thickness changes was mf87171 (row I). Rows A, B, D and possibly G show that increasing OFA sensitivity (OFA Sens) is associated with decreasing retinal thickness (cf. $A^0$ and $A^3$). Decreasing sensitivity was associated with increasing retinal thickness (cf. $I^0$ and $I^3$), see also row G but which seems to show changes in both directions. The row labels at left give the subject ID code and the number of days they were in the study. All subjects except subject mf87171 in row I were DMO patients. Subject mf87171's retinal thickness increased significantly over the 515 days they were in the study, n.b. the green regions of $I^3$.

significant retinal thickening towards the end of the study period ($I^3$, green) that aligned with reduced OFA sensitivity ($I^0$) slopes and shorter response delays ($I^1$, blue). There is reasonable agreement between the locations of OFA change and thickness change, especially if nearest neighbours are included. It is worthwhile emphasizing that the flagged regions in Fig 5 identified with colour mapping, require linear increases or decreases over the study period to reach significance, nonlinear changes may not be flagged. Decreasing retinal thickness corresponded to decreased OFA response delay (Column 2).

To discover more about the eyes that showed significant change in Fig 5 we quantified the differences from normal thickness of the OCT 8×8 thickness grid data (Fig 2A) for the first

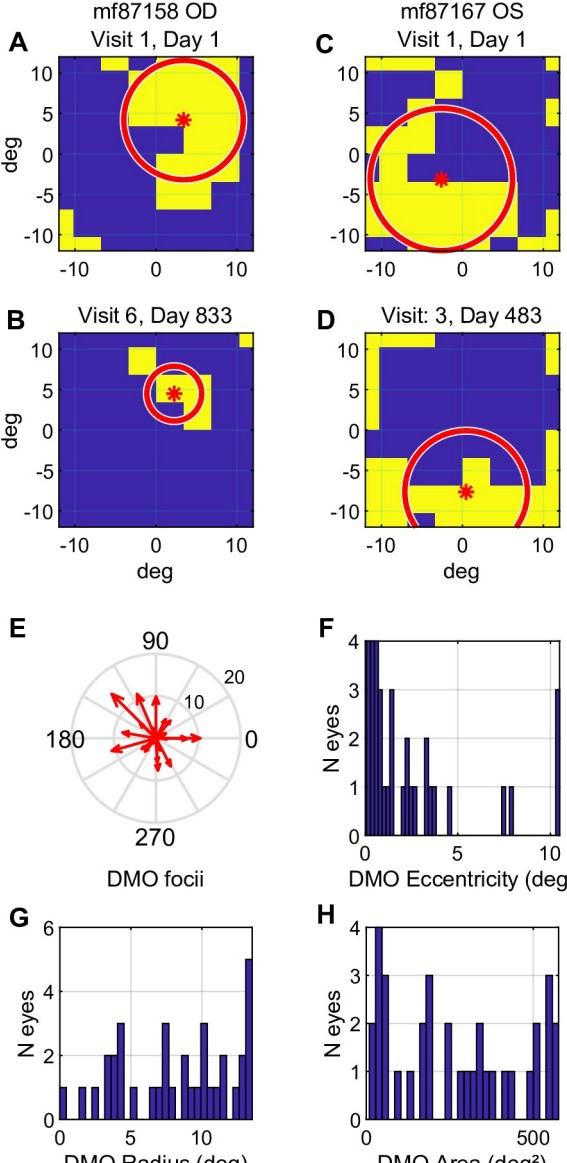

**Fig 6.** A to D are based on the 8×8 OCT thickness grid (Fig 2A). Yellow regions are thicker than normal. The red * is the centroid of the single largest contiguous region of oedema and the red circle is the Equivalent circular region. A,B and C,D are pairs of data sets from the first and last visit for a right and a left eye of two subjects. In these cases the off-axis oedema tends to follow outer radii of the central pit, but never involving the fovea. E. Shows vectors representing the position of the 36 centroids in visual space which can extend beyond 10 degrees eccentricity. F. The same data in Fig 6E, shown as a histogram. G. The radii of the Equivalent circular regions. H. The areas of the single largest contiguous area of oedema from each retina and visit.

and the last visit of each eye. Fig 6A to 6D show examples of the resulting regions of oedema (yellow) presented as left eyes (flipped left-right). We used the Matlab the *regionprops* function to find the single largest contiguous region of oedema, and found its centroid (x,y location), its area and a characteristic radius of the oedematous patch. Fig 6E to 6G summarise the locations and extents of the off-axis oedematous patches. Table 2 shows the centroids and characteristic radii. As shown by Fig 6 the off-axis oedema can partially surround the fovea without involving it. Only 4 eyes had complete foveal involvement on the later visit (mf87172, and mf87171).

**Table 2. Characteristics of oedematous retinal regions.**

| Subject | OD, First | | OD, Last | | OS, First | | OS, Last | |
|---|---|---|---|---|---|---|---|---|
| | x, y | radius | x,y | radius | x,y | radius | x,y | radius |
| mf87157 | 2.57, -0.21 | 8.96 | 0.50, -0.50 | 4.15 | 7.50, -0.10 | +6.56 | 0.50, -0.50 | 4.15 |
| mf87158 | 3.39, 4.18 | 7.38 | 2.25, 4.50 | 3.39 | -10.5, 10.5 | 1.69 | 2.25, 4.50 | 3.39 |
| mf87164 | -1.50, 0.00 | +4.15 | -1.50, 0.00 | +4.15 | 0.00, 9.90 | +5.35 | -1.50, 0.00 | +4.15 |
| mf87167 | 0.27, -5.73 | 7.94 | 3.50, -6.67 | 7.18 | -2.61, -3.17 | 8.79 | 3.50, -6.67 | 7.18 |
| mf87169 | -0.06, 0.94 | 11.7 | 1.50, 0.87 | 11.1 | -0.64, 0.21 | 12.7 | 1.50, 0.87 | 11.1 |
| mf87172 | -0.40, 0.10 | +13.1 | -0.47, -0.22 | +13.2 | -1.12, -0.25 | 12.6 | -0.47, -0.22 | +13.2 |
| mf87174 | -0.77, 2.47 | 10.3 | -0.60, -1.20 | 10.7 | 0.97, -2.91 | 9.87 | -0.60, -1.20 | 10.7 |
| mf87177 | -3.82, 3.53 | 9.42 | -4.50, 10.5 | 3.78 | -7.95, 1.35 | 7.57 | -4.50, 10.5 | 3.78 |
| mf87171 | 2.14, 0.93 | 11.6 | 0.00, 0.00 | +13.5 | 2.33, 1.17 | +10.2 | 0.00, 0.00 | +13.5 |

*The x,y positions of the centroids of the oedema and the characteristic radii as in degree. See text of Fig 6 for an explanation of the steps in the calculations. Radii with a + indicate partial involvement of the fovea.

Fig 7 presents the changes in P60˚ in OFA responses of T2D patients with areas of significant progression or recovery flagged with colour (in the same manner as Fig 5), for the 4 times larger array. Even though the data is often for areas peripheral to the OCT data of Fig 5, the peripheral changes in sensitivity seemed to follow what was happening centrally, cf. mf87157 and mf87171 in Figs 5 and 7.

We next sought to summarise the average significance of the fits to the region-wise slopes reported in Figs 5 and 7. To achieve this, the p-values of the significant slopes were converted to the z-scores of a normal distribution before averaging (and computing SD values) and then were converted back to p-values. This provided asymmetric SD values for the mean p-values (Table 3, -SD and +SD). The procedure produced a statistically robust representation of the relationship between the methods of assessment (Table 3, Mean p) and the number of regions that showed significant slopes of change in response (Table 3, N). In agreement with Figs 5 & 7, Table 3 shows that of the functional measures OFA sensitivity provided the greatest number of significant changes, followed closely by OFA response delays. Similarly, the wide-field stimulus protocol (P60˚) measured the greatest number of significant regional changes in retinal sensitivity, followed by P30˚ delay. The mean p-values were all ≤ 0.021.

For the purposes of averaging the p-values of the significant slopes were converted to z-scores and then back-transformed. The upper half summarise the p-values of Fig 5, the lower Fig 7.

## Discussion

It is now well established that changes in retinal function precede diabetic vasculopathy [17, 18, 35–39]. This aligns closely with neurodegeneration that occurs early in the process of DR, which also proceeds structural changes [40]. The main goal of our work was to extend our understanding of changes in retinal function in at risk eyes with early-DMO. To our knowledge, this is the first report to examine the potential of objective perimetry to identify functional changes in region-wise associations between retinal thickness and multiple measures of functional change measured over-time in patients with early-DMO and good visual acuity. The advantage of mfPOP is that it is rapid and easily administered and objectively measures local retinal function peripheral the fovea complementing BCVA.

DMO is a sight-threatening complication of diabetic retinopathy and the greatest contributor to visual impairment [41]. The traditional standard for assessment of the location of

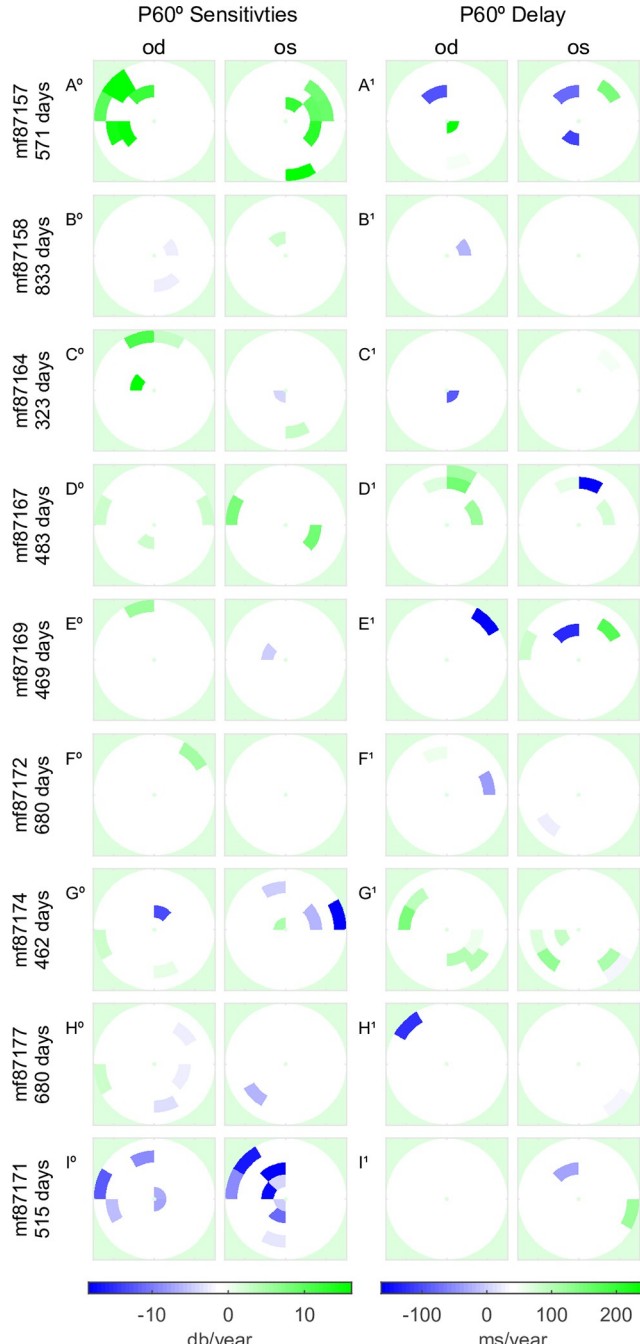

**Fig 7. OFA results presenting the regional radial patterns for the P60˚ stimulus.** Since P60˚ cannot be mapped onto the 10–2 grid it is shown in its native polar layout. Regions of significant linear change are identified with severity according to the legends along the global x-axis. Significant change was also seen at larger retinal eccentricities. There is reasonable concordance between central P60˚ and P30˚, cf. Figs 5 and 7.

abnormalities in the retina involves invasive, time consuming, invasive procedures such as fluorescein angiography [42]. More recently, OCT-angiography has gained prominence for identifying morphological alterations in subjects with diabetes with advantages compared to traditional imaging modalities [43]; however, ocular biomarkers of eyes at risk of progression

**Table 3. Mean and SDs of the significant regions.**

| Fig 5 data | N | Mean p | -SD in p | +SD in p |
|---|---|---|---|---|
| *P30° Sensitivities* | 58 | 0.020 | 0.016 | 0.038 |
| *P30° Delays* | 55 | 0.019 | 0.011 | 0.037 |
| *Matrix Sensitivities* | 5 | 0.017 | 0.013 | 0.022 |
| *OCT Thickness* | 162 | 0.016 | 0.010 | 0.036 |
| Fig 7 data | | | | |
| *P60° Sensitivities* | 53 | 0.021 | 0.014 | 0.043 |
| *P60° Delays* | 39 | 0.020 | 0.014 | 0.038 |

in DMO remain controversial. The validated biomarker that is routinely measured in clinical practice is glycated haemoglobin that identifies a patient's risk of progression in DR. We confirm our previous reports of a peripheral sensitivity loss as a potential surrogate marker of progression in DMO [31]. Figures showing 128 sets of OCT, OFA and Matrix are available as Supplementary material for that paper [31]. In the current clinical guidelines, perimetry is not used in the evaluation of DMO. This may be due to time factor; however, visual acuity is the only functional measure. Newer OFA protocols are shorter in duration and can measure retinal functional more broadly [44]. Considering the value of functional lesions that cannot be measured by imaging or clinical examination, changing to functional measures may be advantageous.

Previous work from our lab has demonstrated the potential of mfPOP to identify eyes that would respond optimally to anti-VEGF treatment in age-related macular degeneration (AMD) [20]. Our additional evaluation of functional changes in diabetic retinopathy has confirmed the utility of mfPOP to identify the severity of non-proliferative DR [18] and the spatial relationship with retinal dysfunction measured by subjective perimetry [31]. Taken together, these findings suggested that it may be possible to detect retinal functional impairment in sight threatening stages of diabetic retinopathy and potentially identify functional biomarkers of eyes at risk of progression. This is all the more important given the possibility of treatments such as fenofibrate having been identified [8, 9]. Here, the presence of sensitivity loss and shorter response delays beyond 4˚ from fixation, identified eyes that increased in retinal thickness overtime (Fig 5). Conversely, eyes that reduced in retinal thickness possibly due to the greater frequency of follow-up and advice, improved in response amplitude, with response delays seeming to be less able to recover from former DMO. The results perhaps indicate that therapeutic interventions might provide recovery of sensitivity.

The importance of examining peripheral function in retinal disease has been confirmed in DR and other inflammatory retinal conditions both from our lab [27] and other groups. Here peripheral areas measured with the P60˚ stimulus seemed to follow central retinal changes in thickness (cf. Figs 5 & 6). Verrotti et al. [45] found progression in DR over an eight year follow-up period in patients who presented at baseline with peripheral sensitivity loss beyond 9˚. Hudson et al. [46] examined retinal function with short-wavelength perimetry in T2D patients with DMO at a time preceding the current advances in retinal imaging. They found that short wavelength perimetry identified greater sensitivity loss than white-on-white perimetry and corresponded more accurately with structural changes in DMO. The loss in sensitivity extended beyond the borders of structural changes; however, lenticular changes often found in diabetes patients was an exclusion criterion in their study. These findings support a shift in assessment guidelines to include functional measures outside the fovea before classical diabetic retinopathy develops and may lead to management strategies that circumvent structural lesions developing.

Long-term follow-up of Matrix sensitivity in our cohort showed little significant change over time and did not reflect changes in retinal structure (Fig 5, Table 3). This may be due to the selective resilience of the magnocellular pathway to inflammatory mediators from surrounding neurons. It is worth noting, however that Matrix is only designed to flag decreased sensitivity. In automated perimetry measuring a sensitivity increase requires near super-human contrast sensitivity, whereas for OFA only a small increase in a large objective response is required. That being said, early functional loss in diabetes has been identified by white-on-white perimetry [45], and short-wavelength perimetry [47, 48], and may represent the impact of early stages of inflammation on outer retinal physiology. Initially confirmed by the work of Feigl et al. [49], pupillary responses activated by both intrinsically photosensitive retinal ganglion cells (ipRGCs) and receptorally by the yellow-ON/blue-OFF cones may have a greater vulnerability to surrounding inflammatory mediators in the retina. In addition, greater than half the input to the pupillary pathway emerges from higher cortical centres [50], which may contribute to neural impairment. Therefore, response delays may represent changes in physiology locally at the retinal level or from higher cortical centres. Evidence that they are afferent defects from the retina has been provided [17].

There is now growing evidence for a bias towards early functional loss in peripheral locations of the visual field [18, 19, 45, 48]. These findings may represent an upregulated response in adjacent healthy tissue in chronic ischemia in the retinal pigmented epithelium and neuroretina more centrally [51].

One of the motivations for this study is the desire to understand when to treat macular oedema given the growing number of effective treatments including dexamethasone [52–57], other steroids, [58] and next generation anti-VEGFs [59]. Further studies on when to treat early-stage DMO with some of these may be aided by information on peripheral visual function.

This study took OCT thickness measurements to be the best indictor of DMO. We used one only one spectral domain device. It is possible that other types of OCT may yield a more accurate assessment such as swept source OCT, or complementary information about the choroid using enhanced depth imaging (EDI). We have recently reported on macular OCT-angiography in young Type 1 diabetics [60] and compared that to an OFA method that tests both eyes in <90 seconds [61]. The rapid OFA test is ideal for children. Future studies of early-stage diabetic eye disease and OFA need to consider comparisons with newer OCT modalities.

Limitations of this study include the number of participants recruited. Comparable to other functional studies in DMO [45, 46], our study represents pilot data that confirms the clinical utility of OFA to measure per-region functional changes in early-DMO over time and may be superior to subjective measures of visual sensitivity. This study was also conducted in Canberra, Australia and may not translate to other countries or ethnic populations. It is important to continue to expand our understanding of functional changes in DMO so that we can develop more detailed recommendations regarding when to start treatment. OFA represents a rapid, objective method of examining functional impairment in DR. The group has recently demonstrated an 82-second version of OFA in AMD [62]. Further studies are needed to confirm and follow-up patients with DMO longitudinally to understand the functional changes during the natural history of inflammation in DMO.

In summary, our study assessed responses to regional multifocal stimuli extending to the central 60 degrees of visual space in eyes with early-DMO and good acuity. We observed that OFA sensitivity declined and response delays become shorter in eyes that increase in retinal thickness over time. Those functional changes were mainly outside the central 4 degrees. Conversely, reductions in retinal thickness showed improvements in response amplitudes, with delays less informative. This model could be used to identify diabetic patients at higher risk of

progression and retinal thickening. In addition, delivery of medications aimed at delaying or reversing the earlier stages of retinopathy [8, 9] may be aligned with functional changes identified towards the borders of the macula.

## Supporting information

**S1 Fig. Multifocal stimulus arrays for the P60˚ and P30˚ stimulus protocols and Illustration of data mapped to the 10–2 format for all 47 data sets.**
(PDF)

**S2 Fig. Shows the OCT retinal thickness data, the P30˚ OFA sensitivity and delay deviations from normal, and the Matrix total deviations, all mapped to the 10–2 grid as per Fig 2 for patient mf87157.** The last two rows show the sensitivity and delay results for the widefield P60˚ stimulus, which cannot be mapped onto the 10–2 grid. Unlike Fig 2 data for the first and last visit are grouped by eye.
(PDF)

## Author Contributions

**Conceptualization:** Ted Maddess.

**Data curation:** Faran Sabeti, Bhim B. Rai, Emilie M. F. Rohan, Richard C. Barry.

**Formal analysis:** Josh P. van Kleef, Corinne F. Carle, Ted Maddess.

**Funding acquisition:** Faran Sabeti, Ted Maddess.

**Supervision:** Faran Sabeti, Ted Maddess.

**Writing – original draft:** Faran Sabeti, Corinne F. Carle.

**Writing – review & editing:** Faran Sabeti, Bhim B. Rai, Josh P. van Kleef, Emilie M. F. Rohan, Richard C. Barry, Rohan W. Essex, Christopher J. Nolan, Ted Maddess.

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
