## [Decision Letter · Decision Letter 0]

19 Jan 2023

PONE-D-22-30047Objective Perimetry identifies regional functional progression and recovery in mild Diabetic Macular OedemaPLOS ONE

Dear Dr. Sabeti,

Thank you for submitting your manuscript to PLOS ONE. After careful consideration, we feel that it has merit but does not fully meet PLOS ONE’s publication criteria as it currently stands. Therefore, we invite you to submit a revised version of the manuscript that addresses the points raised during the review process.

The reviewers have given their insights and comments. Reviewer 2's comments are important and warrants major revision from the authors. The authors need to carefully look at all the issues raised by this reviewer and revise the manuscript appropriately and satisfactorily. Contingent upon the revisions made, it would again needs to reviewed before making a decision.

We look forward to receiving your revised manuscript.

Kind regards,

PremNandhini Satgunam

Academic Editor

PLOS ONE

Journal Requirements:

Reviewers' comments:

Reviewer's Responses to Questions

**Comments to the Author**

1. Is the manuscript technically sound, and do the data support the conclusions?

Reviewer #1: Yes

Reviewer #2: Partly

2. Has the statistical analysis been performed appropriately and rigorously? 

Reviewer #1: Yes

Reviewer #2: Yes

3. Have the authors made all data underlying the findings in their manuscript fully available?

Reviewer #1: Yes

Reviewer #2: Yes

4. Is the manuscript presented in an intelligible fashion and written in standard English?

Reviewer #1: Yes

Reviewer #2: Yes

5. Review Comments to the Author

Reviewer #1: Faran Sabeti et al, al present an interesting and very well written study that evaluates functional progression and recovery in mild Diabetic Macular Oedema

The study results certainly suggest some degrees that Changes in retinal function measured by OFA possibly offer greater power to monitor DMO over time than Matrix perimetry data.

Besides how the magnitude of these data add new findings compare to the current standard cannot be determined based on this study

The results are encouraging and further study is warranted.

here some relevant points :

see the file attachment

Reviewer #2: This study by Sabeti et al presents interesting research findings regarding the comparison of structure function correlation in subjects with DMO. While findings are useful, it is not presented in a very readable manner. Many places assumes that reader would understand, especially the figures or regarding the variation in slopes. More details at specific locations have been requested. The presentation of data is really neat and unique. However, authors make further attempt to clarify more information in the figure legend. Important points have been made in bold.

Abstract:

Please include 1 sentence rationale for the study in the Purpose section.

Methods

What is the time interval between visits.

What is mfPOP variants?

Results

The reduction in retinal thickness seem very small. How is that deemed significant. Which test was used?

What delay are the authors referring to ? It is arbitrary. Numbers should be provided for reduction in retinal thicknesses over time.

“Matrix perimetry quantified many fewer regions that changed significantly.” Need to more specific.

How are conclusions supported by the results? Without providing the result in the abstract and claiming the conclusion does not seem appropriate.

Introduction

Line 3 DR is also functional manifestation?

Line 7 Need to qualify what is fenofibrate?

What is mild “off-centre” DMO?

Lines 14-16 limitations of erg and perimetry should be qualified respectively not grouped together as done presently.

Authors need to justify why not mf ERG which is also objective and gives focal responses? Why mfPOP over mf ERG?

Line 19-21 ; sentence construction needs to be addressed.

Lines 26-29 : how is it relevant in the present context.

The introduction is not flowing well. The exact rationale for the study not coming. Connecting sentences are missing. There are three separate chunks of paragraphs.

Lines 34-40: not suited for this context. Mostly methods related information in the introduction does not fit well.

40-42 :poorly written hypothesis; need to state why you are expecting something.

METHODS;

47-48: not agreeing with statement that both eyes had DMO

More clear information on the how the subjects were recruited needs to be provided.

Lines59-60: Exclusion criteria were absence of XXX conditions seem to suggest the opposite. Please check

Line 62: A set protocol was not followed for each of the patients in terms of the follow-up?

Line 108 : what is “significantly faster or slower than normal”?

Line 138: Expand further on how AmpStd μm to decibel sensitivity (dB)?

Line 191: What was defined as “significant change”?

Correlation heat map of OFA sensitivity changes with OCT thickness changes may be done as done in reference 25

Lines 215 -225: Which points had significant slopes? This section is not clear with respect to the figure. Please expand further.

Line 268-270: “Long-term follow-up of Matrix sensitivity in our cohort showed little significant change over time and did not reflect changes in retinal structure”

Even the OFA sensitivity did not show point-by-point correspondence with change in retinal thicknesses?

What is the implication of delay observed in pupil response?

Please clarify

“In automated perimetry measuring a sensitivity increase requires near super-human contrast sensitivity, whereas for OFA only an increase in a large objective response is required”

Line 276 It is Feigl not Feigle

Why only 10 subjects data presented instead of 16?

Comments on Figures

Fig 1

What are the dashed lines? What does it mean?

Can the rings be clearly demarcated?

Not clear about overlap of stimuli. Please explain clearly.

Panels B and D shows the two halves in each panel are in unequal size. Why is that?

“Reducing the luminance in regions with inherently higher sensitivity enhances the responses in less-responsive regions”

This is not clear and does not belong in the Figure legend.

Mention that there are 11 per each quadrant.

Fig 2

8X8 refers to mm?

Why are the blue panels in panel A tilted?

IMPORTANT: The regions where there is overlap of more than one box; how is the corresponding OCT thickness/OFA parameters considered?

Fig 3

Abstract says 16 patients with baseline DMO but the table 1 and Fig 3 says non-DMO subjects/ Needs fixing

Fig 4

Why this subject data was chosen to present?

Difference map between Visit 4 and Visit 1 may be more useful. Visually, it is difficult to compare so many individual points across the visits.

What is meant by “mapped to the 10-2 format”

What does the square inside represent.? In each of the panels?

What is on the x axis for panels a to d and g to j?

What does Y axis on the left indicate? – 5 and 5 ?

I would keep both visits next to each other for each eye separately for easy comparison?

Line 163 says day 517 and figure says 571

Fig 5

Which slopes are the authors referring to ?

OCT “Thickness” it should be?

What about the remaining 6 subjects?

What is the aim of this figure?

Authors should aim to highlight if retinal thickness and OFA sensitivity region is corresponding to the same points?

Fig 5. In heat map guide what does white mean? No change; it is clearly seen in guide in Fig 6 but not in Fig 5. Please change.

But the heat map does not seem to indicate that?

Why matrix perimetry seems largely unchanged but the retinal thickness changed noticeably in at least 3 subjects?

6. PLOS authors have the option to publish the peer review history of their article (what does this mean?). If published, this will include your full peer review and any attached files.

Reviewer #1: **Yes: **Matias Iglicki MD Certified Teacher of Ophthalmology . International Retinal Group

Reviewer #2: No

---

## [Author Response · Author response to Decision Letter 0]

1 Mar 2023

We are very grateful for the reviews of this manuscript provided by the editor and each of the reviewers. The comments are encouraging and the reviewers appear to share our consideration that this study and its results are clinically important. We appreciate all suggestions from the reviewers, and tried to clarify the purpose and demonstrate some possible limitations of our study. Please see Response to Reviewers document, in black fonts, our detailed response to comments and amendments to manuscript highlighted in red.

---

## [Decision Letter · Decision Letter 1]

11 Apr 2023

PONE-D-22-30047R1Objective Perimetry identifies regional functional progression and recovery in mild Diabetic Macular OedemaPLOS ONE

Dear Dr. Sabeti,

Thank you for submitting your manuscript to PLOS ONE. After careful consideration, we feel that it has merit but does not fully meet PLOS ONE’s publication criteria as it currently stands. Therefore, we invite you to submit a revised version of the manuscript that addresses the points raised during the review process.

ACADEMIC EDITOR:Please look into the comments of Reviewer 2 carefully, and address it systematically. Please also make sure that the changes made are tracked and line numbers are indicated clearly for ease of reviewing.==============================

We look forward to receiving your revised manuscript.

Kind regards,

PremNandhini Satgunam

Academic Editor

PLOS ONE

Additional Editor Comments:

Please make sure the comments are appropriately addressed with proper indication to the changes made. Any sloppiness here will be unfavorable to the review process.

Reviewers' comments:

Reviewer's Responses to Questions

**Comments to the Author**

1. If the authors have adequately addressed your comments raised in a previous round of review and you feel that this manuscript is now acceptable for publication, you may indicate that here to bypass the “Comments to the Author” section, enter your conflict of interest statement in the “Confidential to Editor” section, and submit your "Accept" recommendation.

Reviewer #1: All comments have been addressed

Reviewer #2: (No Response)

2. Is the manuscript technically sound, and do the data support the conclusions?

Reviewer #1: Yes

Reviewer #2: Yes

3. Has the statistical analysis been performed appropriately and rigorously? 

Reviewer #1: Yes

Reviewer #2: Yes

4. Have the authors made all data underlying the findings in their manuscript fully available?

Reviewer #1: Yes

Reviewer #2: Yes

5. Is the manuscript presented in an intelligible fashion and written in standard English?

Reviewer #1: No

Reviewer #2: Yes

6. Review Comments to the Author

Reviewer #1: Authors have addressed every single point properly

every single point has been clarify and improved

Reviewer #2: Thank you for your responses.

Many places in the revised manuscript newly added text are not highlighted in red (for instance line 7-10 and line 37-40) and ideally where does the new text sit in the revised manuscript (in terms of line numbers) needs to be provided. It makes it difficult to identify and track the changes for the reviewer! Authors have misunderstood some of the abstract-related comments and addressed as if it were directed at the body of the paper. Please correct accordingly.

There are still methodological details missing. The results can be improved in terms of ease of reading.

1. Consider rewriting the below sentence

“Subjects were assessed at a median of 3 visits over 457 ± 198 days (mean ± SD) 75 with a median interval of 243.5 ± 145.6 (median ± SD) between visits”

Median is usually supplied with IQR. Mean and median used interchangeably. Also mention in the abstract.

“We enrolled T2D patients with mild DMO, and similar T2D patients without DMO at baseline, and

followed them longitudinally for 1.5 years.”

A version of this statement needs to be in abstract methods

2. Expand mfpop and OFA in abstract methods

3. “Matrix perimetry quantified many fewer regions that changed significantly”

Which regions need to be mentioned at least grossly

4. Authors need to justify why not mf ERG which is also objective and gives focal responses?

Why mfPOP over mf ERG? The justification is not sufficient. Is any distinct advantage?

5. Line 50 need to provide references for previous findings

6. P 30 and P 60 which one is subtending 15 and 30 degrees? Please clarify.

7. “many fewer” in the abstract results sounds paradoxical. Please change.

Lines 109-110

8. “Total stimulus duration was 6 minutes divided into 9 segments of 40s with short breaks between segments”

Assuming short break is 10 seconds; close to additional 1 min and 20 seconds is added to the protocol. Therefore the total testing time should be mentioned as 7 to 8 minutes.

Methods:

Important points

Spatial extent of DMO (in degrees) in each of the patients’ needs to be quantified at baseline and the follow up visit. Preferably as table.

OFF centre DMO – how much was it off? How many degrees away from the macula? Need to be quantified.

This has direct implications on the structure-function mapping. That can help in understand how some points recover and not others.

Fig 4 suggestion

Keep OD basliene and follow up together instead of OD and OS together. Applies to OS as well,

If the DMO is well restricted within the macula, why are the peripheral points showing improvement or deterioration? In Fig 5 and 6

 

Fig 4 legend says it is Diabetic patient but the patient is DMO patient. Needs to be changed

Fig 5

Each parameter increase or decrease may mean deterioration or recovery

Increase in sensitivity is improvement but increase in thickness is perhaps worsening

So one suggestion is to add alphabet R (recovery) or W (worsening) at either end of the legend for each variable to help the reader. In fact letter choice is up to the authors

Difference is taken baseline – last visit or vice versa?. This information needs to be present in both Fig 5 and 6 legend.

Discussion point

What are the implications of delays?

What mechanisms is responsible for the delays? Can you point to me if it has been already mentioned explicitly in the paper?

7. PLOS authors have the option to publish the peer review history of their article (what does this mean?). If published, this will include your full peer review and any attached files.

Reviewer #1: **Yes: **Matias Iglicki MD

Reviewer #2: No

---

## [Author Response · Author response to Decision Letter 1]

13 Apr 2023

Reviewer Comments: 

Reviewer #1: All comments have been addressed.

Reviewer #2:

Many places in the revised manuscript newly added text are not highlighted in red (for instance line 7-10 and line 37-40) and ideally where does the new text sit in the revised manuscript (in terms of line numbers) needs to be provided. It makes it difficult to identify and track the changes for the reviewer! Authors have misunderstood some of the abstract-related comments and addressed as if it were directed at the body of the paper. Please correct accordingly.

There are still methodological details missing. The results can be improved in terms of ease of reading.

Thank you for pointing this out. We apologise that we did not rigorously check that the additions were highlighted prior to our last revised submission. We have now reviewed the amendments following the reviewer’s comments and correctly identified them in RED if original and GREEN if secondary revisions apply for this round. 

1. Consider rewriting the below sentence

“Subjects were assessed at a median of 3 visits over 457 ± 198 days (mean ± SD) 75 with a median interval of 243.5 ± 145.6 (median ± SD) between visits”

Median is usually supplied with IQR. Mean and median used interchangeably. Also mention in the abstract.

“We enrolled T2D patients with mild DMO, and similar T2D patients without DMO at baseline, and

followed them longitudinally for 1.5 years.”

A version of this statement needs to be in abstract methods

We have now amended the abstract to read:

“We did that longitudinally in Type 2 diabetes (T2D) patients with mild Diabetic Macular Oedema (DMO) with good vision and a similar number of T2D patients without DMO”

2. Expand mfpop and OFA in abstract methods

Addressed, thank you.

3. “Matrix perimetry quantified many fewer regions that changed significantly”

Which regions need to be mentioned at least grossly

We have now added the following statement to the Abstract: “mostly presenting in the central 8 degrees.”

4. Authors need to justify why not mf ERG which is also objective and gives focal responses?

Why mfPOP over mf ERG? The justification is not sufficient. Is any distinct advantage?

Thank you for pointing this out. We have now added the following statement to the introduction

“In addition, OFA requires minimal training to administer with rapid setup times.”

5. Line 50 need to provide references for previous findings

Addressed, thank you.

6. P 30 and P 60 which one is subtending 15 and 30 degrees? Please clarify.

We have now added the following statement to the Methods:

“We refer to the two variants as P30° and P60° pertaining to the ±15° and ± 30° stimulus array, respectively”

7. “many fewer” in the abstract results sounds paradoxical. Please change.

We have now amended the text in the abstract.

Lines 109-110

8. “Total stimulus duration was 6 minutes divided into 9 segments of 40s with short breaks between segments”

Assuming short break is 10 seconds; close to additional 1 min and 20 seconds is added to the protocol. Therefore the total testing time should be mentioned as 7 to 8 minutes.

Corrected, thank you for pointing this out.

Methods:

Important points

Spatial extent of DMO (in degrees) in each of the patients’ needs to be quantified at baseline and the follow up visit. Preferably as table.

OFF centre DMO – how much was it off? How many degrees away from the macula? Need to be quantified.

This has direct implications on the structure-function mapping. That can help in understand how some points recover and not others.

Fig 4 suggestion

Keep OD baseline and follow up together instead of OD and OS together. Applies to OS as well,

We agree the suggested presentation method has some merit scientifically. As mentioned before the current presentation method conforms to standard ophthalmic practice for OU data sets with OD on the left, as the patient presents to the clinician. Fig 4 is presented to illustrate the format of the data more than to quantify anything. In particular, the current format sets up understanding of the format of Figure 5, which presents the significant slopes in OD/OS format. Flipping the columns to baseline/follow-up would mean that feature of the flow of the figures, and so too understanding, would be lost. Quantification is done for all the data Figures 5 and 6 (also in OD/OS format).

Another nice feature of the original Fig. 4 is that it allows changes to be seen by free-fusion of the pairs of baseline and follow-up columns as if they are binocular pairs. This makes it immediately obvious what is different. We now mention this in the Figure legend saying

“Readers may wish to free-fuse A and B”

That is also true of all 47 data sets in Supplementary Figure 1. This feature is lost if the columns are swapped as suggested. That being said the idea has merit, therefore we have done as suggested in Supplementary Figure 2. At the end of the paragraph on Figure 4 we now say

“The same data is presented with the baseline data for both eyes on the left in Supplementary Figure 2.”

If the DMO is well restricted within the macula, why are the peripheral points showing improvement or deterioration? In Fig 5 and 6

We have postulated on the origin of changes in retinal function in the discussion (lines 308 – 318). We now also add that structural changes may have been due to the frequency of clinical review and advice from the attending physician for diet and lifestyle modification. We have now added the following sentence on line 286 of the discussion:

“Conversely, eyes that reduced in retinal thickness possibly due to the greater frequency of follow-up and advice, improved in response amplitude, with response delays seeming to be less able to recover from former DMO. “  

Fig 4 legend says it is Diabetic patient but the patient is DMO patient. Needs to be changed

Addressed, thank you.

Fig 5

Each parameter increase or decrease may mean deterioration or recovery

Increase in sensitivity is improvement but increase in thickness is perhaps worsening

So one suggestion is to add alphabet R (recovery) or W (worsening) at either end of the legend for each variable to help the reader. In fact letter choice is up to the authors

Difference is taken baseline – last visit or vice versa?. This information needs to be present in both Fig 5 and 6 legend.

We did not look at change as the difference between the first and last visit. Firstly, that would throw away most of the data from the many intermediate visits, which are shown in Figure 3. Another reason for not taking the difference between the first and last visit is that the duration of time between those visits is different for every subject (Fig. 3), so the differences would not be calibrated for a uniform time period. To resolve both issues we fitted a line to the data to compute the slopes in dB/year, ms/year and um/year. That process also meant we could understand if the slopes were significant or not. The colour-coded slopes are reported with those units at the bottom of Figures 5 and 6. This was made clearer by amending the last sentence of the Methods which now reads

“We used multiple regression-linear models to examine the changes in retinal thickness, and changes in sensitivity and response delay overtime (see below). Thus, significant change was quantified by slopes with p-values < 0.05.”

The word slope then appears 8 more times in the text and there is no mentioned of computing the difference between the first and last visits. The text for Figure 5 says

“The longitudinal data over time were fitted by linear regression and those regions with significant change in retinal sensitivity or thickness over time between baseline and final visit are coloured. The colour-map legends at the bottom of the pairs of columns indicate the magnitude and sign of those per-region slopes.”

The main point about the observed slopes for OFA is that they appear to follow changes in retinal thickness, whether retinal thickness improved or got worse. This is detailed on page 16.

“Here, the presence of sensitivity loss and shorter response delays beyond 4° from fixation, identified eyes that increased in retinal thickness overtime (Fig. 5). Conversely, eyes that reduced in retinal thickness possibly due to the greater frequency of follow-up and advice, improved in response amplitude, with response delays seeming to be less able to recover from former DMO. The results perhaps indicate that therapeutic interventions might provide recovery of sensitivity.”

The reviewer’s comments in green helped clarify this. Thank you.

Discussion point

What are the implications of delays?

What mechanisms is responsible for the delays? Can you point to me if it has been already mentioned explicitly in the paper?

We hypothesize that response delays measured by OFA may represent neural impairment from both local retinal inflammation or higher cortical neuropathy. We have added the following sentence in the discussion:

“Therefore, response delays may represent changes in physiology locally at the retinal level or from higher cortical centres.“ 

It is important to note that the observed changes are not due to iris neuropathy which would create global shifts of delay that were the same for every region. Instead, we see in the 47 data sets of Supplementary Figure 1 many cases where slower and faster than normal delays occur in the same retina. Therefore, it appears that these are local functional defects. The fact that they are afferent defects from the retina was demonstrated by reference 17 (n.b. its title).

So, we have now added to the above at line 317

“Evidence that they are afferent defects from the retina has been provided (17).”

We thank the reviewer for allowing us to make these valuable points.

---

## [Decision Letter · Decision Letter 2]

8 May 2023

PONE-D-22-30047R2Objective Perimetry identifies regional functional progression and recovery in mild Diabetic Macular OedemaPLOS ONE

Dear Dr. Sabeti,

Thank you for submitting your manuscript to PLOS ONE. After careful consideration, we feel that it has merit but does not fully meet PLOS ONE’s publication criteria as it currently stands. Therefore, we invite you to submit a revised version of the manuscript that addresses the points raised during the review process.

We look forward to receiving your revised manuscript.

Kind regards,

PremNandhini Satgunam

Academic Editor

PLOS ONE

Journal Requirements:

Additional Editor Comments:

Thanks for the revision. There still remains few minor pending revisions. I would urge you to pay careful attention to this. This would enhance this work.

Reviewers' comments:

Reviewer's Responses to Questions

**Comments to the Author**

1. If the authors have adequately addressed your comments raised in a previous round of review and you feel that this manuscript is now acceptable for publication, you may indicate that here to bypass the “Comments to the Author” section, enter your conflict of interest statement in the “Confidential to Editor” section, and submit your "Accept" recommendation.

Reviewer #2: (No Response)

2. Is the manuscript technically sound, and do the data support the conclusions?

Reviewer #2: Yes

3. Has the statistical analysis been performed appropriately and rigorously? 

Reviewer #2: Yes

4. Have the authors made all data underlying the findings in their manuscript fully available?

Reviewer #2: Yes

5. Is the manuscript presented in an intelligible fashion and written in standard English?

Reviewer #2: Yes

6. Review Comments to the Author

Reviewer #2: Other comments have been addressed. However, these comments are have not been addressed.

Spatial extent of DMO (in degrees) in each of the patients’ needs to be quantified at baseline and the follow up visit. Preferably as table.

OFF centre DMO – how much was it off? How many degrees away from the macula? Need to be quantified.

This has direct implications on the structure-function mapping. That can help in understand how some points recover and not others.

7. PLOS authors have the option to publish the peer review history of their article (what does this mean?). If published, this will include your full peer review and any attached files.

Reviewer #2: No

---

## [Author Response · Author response to Decision Letter 2]

15 May 2023

Thank you for this comment we agree this type of information could potentially add to understanding in terms of structure function mapping and what is going on with recovery and progression. We therefore measured the distribution of DMO in each eye quantifying:- 

1) The area of the main oedematous region

2) The x,y coordinates of the centroid of that region in each eye

This was done for the first and last visit of each from those subjects who showed a significant change in DMO over the study in at least one eye, i.e. those subjects of Figure 5. That provided 36 eyes (2 visits * 18 eyes) for the analysis. We used the OCT 8×8 thickness grid (Fig. 2A) data from those eyes. Before analysis we flipped the data from right eyes to match left eyes. We then formed an 8×8 normative template by taking the median at each of the 8×8 locations for the 36 eyes. The template was thus the typical 8×8 thickness profile with outliers removed. The template was then subtracted from each 8×8 data set. Regions of the retinas that were thicker than normal were considered oedematous. We used the regionprops function of Matlab to identify the areas of oedema from each retina and found: 1) the area of the largest single region/eye, 2) its Equivalent Radius, and the x,y coordinates of its Centroid. We agree with the reviewer that it is sensible to project the data into visual space to match Figure 5. The new Table 3 represents the requested data. 

The steps of the analysis behind Table 3 and some summary statistics are provided in the new Figure 6. The old Figure 6 is now Figure 7. Figure 6 and its legend and the accompanying text are provided here.

 OD, First OD, Last OS, First OS, Last

Subject x, y radius x,y radius x,y radius x,y radius

mf87157 2.57, -0.21 8.96 0.50, -0.50 4.15 7.50, -0.10 ˖6.56 0.50, -0.50 4.15

mf87158 3.39, 4.18 7.38 2.25, 4.50 3.39 -10.5, 10.5 1.69 2.25, 4.50 3.39

mf87164 -1.50, 0.00 ˖4.15 -1.50, 0.00 ˖4.15 0.00, 9.90 ˖5.35 -1.50, 0.00 ˖4.15

mf87167 0.27, -5.73 7.94 3.50, -6.67 7.18 -2.61, -3.17 8.79 3.50, -6.67 7.18

mf87169 -0.06, 0.94 11.7 1.50, 0.87 11.1 -0.64, 0.21 12.7 1.50, 0.87 11.1

mf87172 -0.40, 0.10 ˖13.1 -0.47, -0.22 ˖13.2 -1.12, -0.25 12.6 -0.47, -0.22 ˖13.2

mf87174 -0.77, 2.47 10.3 -0.60, -1.20 10.7 0.97, -2.91 9.87 -0.60, -1.20 10.7

mf87177 -3.82, 3.53 9.42 -4.50, 10.5 3.78 -7.95, 1.35 7.57 -4.50, 10.5 3.78

mf87171 2.14, 0.93 11.6 0.00, 0.00 ˖13.5 2.33, 1.17 ˖10.2 0.00, 0.00 ˖13.5

*The x,y positions of the centroids of the oedema and the characteristic radii as in degree. See text of Fig. 6 for an explanation of the steps in the calculations. Radii with a ˖ indicate partial involvement of the fovea.

Figure 6. A to D are based on the 8×8 OCT thickness grid (Fig. 2A). Yellow regions are thicker than normal. The red * is the centroid of the single largest contiguous region of oedema and the red circle is the Equivalent circular region. A,B and C,D are pairs of data sets from the first and last visit for a right and a left eye of two subjects. In these cases the off-axis oedema tends to follow outer radii of the central pit, but never involving the fovea. E. Shows vectors representing the position of the 36 centroids in visual space which extend beyond 10 degrees eccentricity. F. The same data shown as a histogram. G. The radii of the Equivalent circular regions. H. The areas of the single largest contiguous area of oedema from each retina and visit.

We also added the following text to the Results which now reads:

“To discover more about the eyes that showed significant change in Figure 5 we quantified the differences from normal thickness of the OCT 8×8 thickness grid data (Fig. 2A) for the first and the last visit of each eye. Figure 6 A to D show examples of the resulting regions of oedema (yellow). We used the Matlab the regionprops function to find the single largest contiguous region of oedema, and found its centroid (x,y location), its area and a characteristic radius of the oedematous patch. Fig. 6E to G summarise the locations and extents of the off-axis oedematous patches. Table 3 shows the centroids and characteristic radii. As shown by Fig. 6 the off-axis oedema can partially surround the fovea without involving it. Only 4 eyes had complete foveal involvement on the later visit (mf87172, and mf87171).”

---

## [Decision Letter · Decision Letter 3]

29 May 2023

PONE-D-22-30047R3Objective Perimetry identifies regional functional progression and recovery in mild Diabetic Macular OedemaPLOS ONE

Dear Dr. Sabeti,

Thank you for submitting your manuscript to PLOS ONE. After careful consideration, we feel that it has merit but does not fully meet PLOS ONE’s publication criteria as it currently stands. Therefore, we invite you to submit a revised version of the manuscript that addresses the points raised during the review process.

ACADEMIC EDITOR: please see the comments below

We look forward to receiving your revised manuscript.

Kind regards,

PremNandhini Satgunam

Academic Editor

PLOS ONE

Journal Requirements:

Additional Editor Comments:

The reviewer has given a careful look, and there are still some lingering issues that needs to be fixed. I urge the authors to be accurate and pay careful attention to the comments, and fix the ambiguity that the reviewer has raised.

Reviewers' comments:

Reviewer's Responses to Questions

**Comments to the Author**

1. If the authors have adequately addressed your comments raised in a previous round of review and you feel that this manuscript is now acceptable for publication, you may indicate that here to bypass the “Comments to the Author” section, enter your conflict of interest statement in the “Confidential to Editor” section, and submit your "Accept" recommendation.

Reviewer #2: All comments have been addressed

2. Is the manuscript technically sound, and do the data support the conclusions?

Reviewer #2: Yes

3. Has the statistical analysis been performed appropriately and rigorously? 

Reviewer #2: (No Response)

4. Have the authors made all data underlying the findings in their manuscript fully available?

Reviewer #2: Yes

5. Is the manuscript presented in an intelligible fashion and written in standard English?

Reviewer #2: Yes

6. Review Comments to the Author

Reviewer #2: I am pleased that authors have taken efforts to address these key points.

The new panels A and B (Fig 6) shows the changes in oedema seem like a mirror image of change in OCT retinal thickness in the left most panel (Fig 5). This is probably due to flipping of data to represent the corresponding region in visual space.

Please confirm that if the data in Fig 6 is indeed in retinal space.

Therefore, add a note in Fig 6 legend that if the data is presented here in retinal space.

Panel E legend not accurate. It shows all centroids not just those greater than 10 degrees

7. PLOS authors have the option to publish the peer review history of their article (what does this mean?). If published, this will include your full peer review and any attached files.

Reviewer #2: No

---

## [Author Response · Author response to Decision Letter 3]

31 May 2023

Thank you for this comment. No, Figure 6 A-D are presented as left eyes and not presenting as corresponding regions in retinal space. They were flipped left right so that we could measure the median values across all eyes. It is also worth noting the Fig 5B3 is derived from the fitted slopes not the edema compared to normative data on the first and last days, like Fig 6, so they would not be expected to be similar. To clarify this in the text we have added the following statement to the Results: 

“To discover more about the eyes that showed significant change in Figure 5 we quantified the differences from normal thickness of the OCT 8×8 thickness grid data (Fig. 2A) for the first and the last visit of each eye. Figure 6 A to D show examples of the resulting regions of oedema (yellow) presented as left eyes (flipped left-right). We used the Matlab the regionprops function to find the single largest contiguous region of oedema, and found its centroid (x,y location), its area and a characteristic radius of the oedematous patch. Fig. 6E to G summarise the locations and extents of the off-axis oedematous patches. Table 3 shows the centroids and characteristic radii. As shown by Fig. 6 the off-axis oedema can partially surround the fovea without involving it. Only 4 eyes had complete foveal involvement on the later visit (mf87172, and mf87171).

Figure 6E. presents all the centroids but to enhance visualisation of the central ones we presented the same data as a histogram in Fig 6F. We had confused the situation by leaving out the word “can”, which is now inserted. To make this clearer in the text we have modified the Fig 6. Legends to read:

Fig 6. A to D are based on the 8×8 OCT thickness grid (Fig. 2A). Yellow regions are thicker than normal. The red * is the centroid of the single largest contiguous region of oedema and the red circle is the Equivalent circular region. A,B and C,D are pairs of data sets from the first and last visit for a right and a left eye of two subjects. In these cases the off-axis oedema tends to follow outer radii of the central pit, but never involving the fovea. E. Shows vectors representing the position of the 36 centroids in visual space which can extend beyond 10 degrees eccentricity. F. The same data in Fig. 6E, shown as a histogram. G. The radii of the Equivalent circular regions. H. The areas of the single largest contiguous area of oedema from each retina and visit.

---

## [Editor Report · Decision Letter 4]

4 Jun 2023

Objective Perimetry identifies regional functional progression and recovery in mild Diabetic Macular Oedema

PONE-D-22-30047R4

Dear Dr. Sabeti,

We’re pleased to inform you that your manuscript has been judged scientifically suitable for publication and will be formally accepted for publication once it meets all outstanding technical requirements.

Kind regards,

PremNandhini Satgunam

Academic Editor

PLOS ONE

Additional Editor Comments (optional):

Thanks for clarifying the concerns
---

## [Editor Report · Acceptance letter]

8 Jun 2023

PONE-D-22-30047R4 

Objective Perimetry identifies regional functional progression and recovery in mild Diabetic Macular Oedema 

Dear Dr. Sabeti:

I'm pleased to inform you that your manuscript has been deemed suitable for publication in PLOS ONE. Congratulations! Your manuscript is now with our production department. 

Kind regards, 

on behalf of

Dr. PremNandhini Satgunam 

Academic Editor

PLOS ONE